# Pro-Resolving Mediators in Rotator Cuff Disease: How Is the Bursa Involved?

**DOI:** 10.3390/cells13010017

**Published:** 2023-12-20

**Authors:** Franka Klatte-Schulz, Nicole Bormann, Aysha Bonell, Jasmin Al-Michref, Hoang Le Nguyen, Pascal Klöckner, Kathi Thiele, Philipp Moroder, Martina Seifert, Birgit Sawitzki, Britt Wildemann, Georg N. Duda

**Affiliations:** 1Julius Wolff Institut, Berlin Institute of Health at Charité-Universitätsmedizin Berlin, 13353 Berlin, Germany; 2BIH-Center for Regenerative Therapies, Berlin Institute of Health at Charité-Universitätsmedizin Berlin, 13353 Berlin, Germany; 3Center for Musculoskeletal Surgery, Charité-Universitätsmedizin Berlin, 13353 Berlin, Germany; 4Vivantes Auguste Viktoria Klinikum, 12157 Berlin, Germany; 5Schulthess Klinik, 8008 Zurich, Switzerland; 6Institute of Medical Immunology, Charité-Universitätsmedizin Berlin, Corporate Member of Freie Universität Berlin and Humboldt University of Berlin, 13353 Berlin, Germany; 7Center of Immunomics, Berlin Institute of Health at Charité-Universitätsmedizin Berlin, 13353 Berlin, Germany; 8Experimental Trauma Surgery, Department of Trauma-, Hand- and Reconstructive Surgery, Jena University Hospital, Friedrich Schiller University Jena, 07747 Jena, Germany

**Keywords:** subacromial bursa, rotator cuff disease, pro-resolving mediators, resolution of inflammation, mechanical stress/loading

## Abstract

So far, tendon regeneration has mainly been analyzed independent from its adjacent tissues. However, the subacromial bursa in particular appears to influence the local inflammatory milieu in the shoulder. The resolution of local inflammation in the shoulder tissues is essential for tendon regeneration, and specialized pro-resolving mediators (SPMs) play a key role in regulating the resolution of inflammation. Here, we aimed to understand the influence of the bursa on disease-associated processes in neighboring tendon healing. Bursa tissue and bursa-derived cells from patients with intact, moderate and severe rotator cuff disease were investigated for the presence of pro-resolving and inflammatory mediators, as well as their effect on tenocytes and sensitivity to mechanical loading by altering SPM signaling mediators in bursa cells. SPM signal mediators were present in the bursae and altered depending on the severity of rotator cuff disease. SPMs were particularly released from the bursal tissue of patients with rotator cuff disease, and the addition of bursa-released factors to IL-1β-challenged tenocytes improved tenocyte characteristics. In addition, mechanical loading modulated pro-resolving processes in bursa cells. In particular, pathological high loading (8% strain) increased the expression and secretion of SPM signaling mediators. Overall, this study confirms the importance of bursae in regulating inflammatory processes in adjacent rotator cuff tendons.

## 1. Introduction

The treatment of tendon-associated diseases represents one of the remaining unsolved clinical problems in orthopedics, with a constantly growing number of patients suffering and no solutions to regenerate injured tendons being available. Specifically, shoulder tendon regeneration is a complex process, which, so far, has been seen to occur mainly independently from surrounding tissues. More recently, we and others hypothesized that adjacent tissues substantially impact the local tendon regenerative cascade, and due to its close localization, specifically the subacromial bursa near the rotator cuff is thought to impact the local inflammatory milieu [1,2,3].

Bursae are synovial structures functioning as friction-reducing cushions in articulating joints. Consequently, these bursa tissues are in locations that experience high mechanical loading [4]. The largest bursa in the human body is the subacromial bursa located in the shoulder between the acromion and the rotator cuff tendons. This anatomical location underlines the clinical relevance of the bursa to degenerative and regenerative processes in the shoulder, as well as their association with shoulder pain [5]. Shoulder pain is amongst others caused by subacromial bursitis, an inflammation of the subacromial bursa [6,7,8]. Bursitis is often treated conservatively using physiotherapy or an injection of anti-inflammatory agents, whereas in severe cases that do not respond to conservative treatments, the inflamed bursa tissue is removed (bursectomy) [9]. This procedure is controversially discussed as meaningful therapy since the subacromial bursa is also thought to be essential to healing: The bursa contains a relevant pool of progenitor cells [10,11,12,13], and with its tight fibrovascular network, it covers the tendon and enables initial tendon repair processes [4,14]. Indeed, immune cells and pro-inflammatory cytokines play an important role not only in bursitis but also in healthy bursae [15]. In particular, inflammation-associated markers, such as stromal cell-derived factor 1 (SDF-1), Interleukin 1β (IL-1β), IL-6, tumor necrosis factor α (TNF-α) and transforming growth factor β (TGF-β), as well as matrix metalloproteinases (MMPs) and pain-associated mediators cyclooxygenase 1 and 1 (COX-1/-2), are upregulated in the subacromial bursae of patients with a rotator cuff tear compared to healthy controls [16,17,18,19,20,21]. It can be speculated on the one hand that pro-inflammatory and pain-generating mediators stimulate tendinopathic processes or on the other hand that the bursa can serve as a reservoir for essential inflammatory cells and healing-promoting mediators that are able to initiate and promote tendon tissue repair.

To overcome chronic inflammation in the bursa, the timely resolution of inflammation would be required to restore tissue homeostasis. The resolution of inflammation is an active process that is regulated by specialized pro-resolving mediators (SPMs). These SPMs include lipoxins, resolvins, protectins and maresins, which are synthesized from omega-3 essential fatty acids via an enzymatic reaction in response to tissue injury [22]. These SPMs bind to their specific SPM receptors, such as the formyl peptide receptor (FPR), Chemerin Receptor 23 (ChemR23) or G-protein-coupled receptor 18 or 23 (GPR18/23), which are found on a large variety of cells. SPMs act upon their target cells by inhibiting the migration of pro-inflammatory cells, such as neutrophiles, lymphocytes and M1 macrophages, into inflamed tissues, and they also inhibit their expression and release of pro-inflammatory factors [22,23]. Additionally, Annexin A1 (ANXA1), which is not a typical SPM but a pro-resolving mediator, binds to the SPM receptor FPR2 and thus also initiates a pro-resolving response [24,25]. For the rotator cuff, it has been shown that the SPM signaling mediators FPR2 and ChemR23 are expressed in higher amounts in early-stage diseases in patients receiving subacromial decompression than in advanced-stage diseases in patients with a full-thickness SSP tear [26]. The contribution of pro-resolving mediators in the attempt to counteract pro-inflammatory processes has recently been confirmed in a rat overuse tendon model [27]. In vitro, the therapeutic potential of SPMs in promoting the resolution of inflammation in tendon healing has been illustrated with lipoxin B_4_ (LXB_4_) and resolvin E1 (RvE1), as well as the aspirin-triggered lipoxin isoform 15-epi LXA_4_, preventing inflammatory processes in IL-1β- or LPS-treated patient-derived tendon cells [26,28]. In contrast to anti-inflammatory drugs used for dampening prolonged inflammatory processes, which reduces pro-resolving mediators in addition to inflammatory factors [29], SPMs appear promising as a targeted therapeutic intervention to overcome compromised tendon healing and maintain tendon homeostasis.

However, an understanding of the role of the subacromial bursa in pro-resolving processes in tendon pathologies, particularly the rotator cuff tendons, is missing. We hypothesize that bursae home pro-resolving mediators capable of modulating inflammatory processes at the nearby tendon rupture side. This process depends on the severity of tendon disease and is regulated by mechanical loading. To verify this hypothesis, we unraveled the role of SPMs and their receptors in bursa tissue from patients with different severities of rotator cuff disease, ranging from an intact rotator cuff to moderate rotator cuff disease (patients with impingement or partial tear) and severe rotator cuff disease (patients with a full-thickness tear). To this end, we (1) identified SPM signaling mediators, immune cells and components of inflammatory pathways in bursa tissues; (2) determined the pro-resolving effect of bursa-released factors on tenocytes; and (3) investigated the role of mechanical loads on SPM signaling mediators in bursa cells to understand possible regulatory mechanisms. The study design is depicted in Figure 1.

## 2. Materials and Methods

### 2.1. Tissue Harvesting

To achieve the study objectives, subacromial bursa samples from patients with different severities of rotator cuff disease were used to understand regulations of pro-resolving pathways in the bursa–tendon interplay.

Subacromial bursa samples were harvested from the lateral subacromial site during shoulder surgery from patients who gave their written informed consent prior to the study. Bursae from patients with a macroscopically intact rotator cuff (shoulder or acromioclavicular (AC) joint instability) were taken during minimally invasive or open surgery and considered the intact group. Patients with a degenerative partial tear of the SSP or with isolated impingement syndrome were considered the moderate rotator cuff disease group, and patients with a degenerative full-thickness tear of the SSP were included as the severe rotator cuff disease group. Samples from the SSP disease groups were taken during minimally invasive arthroscopic surgery. Additionally, SSP tendon samples were collected from patients with a full-thickness SSP tear and used for the isolation of tenocytes. Tear morphology was evaluated using the Patte classification for the extent of tendon retraction [30], and, intraoperatively, the tear size was classified according to Bayne and Bateman [31]. Partial tears were classified according to Snyder [32]. Patient demographics are summarized in Table 1. Depending on the size of the biopsies, they were separated for different analyses. The n-number varied according to the performed analyses. The study was approved by the Charité institutional review board (EA1/267/15).

### 2.2. Identification of Pro-Resolving and Inflammatory Mediators in Bursa Tissue at RNA Level

In the first step, we aimed to identify the pro-resolving and inflammatory mediators in the bursa tissue of the three disease groups to understand the relationship between the inflammatory and resolving status of the bursa and the disease progression of the rotator cuff tendon tears.

After tissue harvesting, the bursa samples were subsequently frozen in sampling tubes in liquid nitrogen and stored at −80 °C until RNA isolation. RNA isolation was performed as described previously [15,33]. The frozen bursa tissue was homogenized using a liquid-nitrogen-cooled steel mortar system and peqGOLD Trifast™ (Peqlab, Erlangen, Germany). Chloroform was used for phase separation, and RNA was purified using a NucleoSpin^®^ RNA kit (Macherey-Nagel, Düren, Germany). The purity and quantity of the RNA was analyzed with a Nanodrop ND1000 system (Peqlab). A total of 100 ng RNA was transcribed into cDNA with the qScript^®^ cDNA Supermix (Quanta Biosciences, Beverly; MA, USA). The expression levels of SPM signaling molecules, as well as those of the pro- and anti-inflammatory cytokines of the interferon-γ (IFN-γ), nuclear factor ‘kappa-light-chain-enhancer’ of activated B cells (NF-κB), glucocorticoid and signal transducer and activator of transcription 6 (STAT6) pathways, were evaluated (Table 2). qRT-PCR was performed with the PerfeCTa^®^ SYBR^®^ Green SuperMix (Quanta Biosciences) according to the manufacturer’s manual and a LightCycler 480 System (Roche, Basel, Switzerland). Primers were designed using Primer 3 software (https://primer3.ut.ee/ (accessed on 6 January 2020)) or sequences adapted from the literature [26,34]. The primer sequences are depicted in Appendix A, and the primers were produced by Tib MolBiol, Berlin, Germany. All primers were tested for amplification efficiency, and an efficiency correction equation was used to calculate the normalized expression to the three reference genes Ppia, HPRT and 18s.

### 2.3. Multiplex Immunofluorescence Staining of SPM Receptors

To verify the findings at the RNA level and to better understand the spatial distribution of SPM receptors in the bursa tissue and their co-expression with immune cells, multiplex immunofluorescence staining was performed.

Therefore, harvested bursa tissue was fixed in 4% PFA for 24 h and embedded in paraffin as described previously [15]. For multiplex staining, 4 μm thick sections were stained with the primary antibodies for the SPM receptors ChemR23 and FPR2 and the immune cell marker CD45. Prior to staining, the slices were pre-treated in Tris/EDTA buffer (pH 9) at 120 °C with 1.9 bar in a pressure cooker for 3 min. Afterwards, the slices were incubated in Tris-buffered saline solution (TBS) with 0.05% Tween 20 two times for 5 min each. Blocking was performed for 1 h with 10% normal goat serum (NGS) in 1% BSA/TBS solution at room temperature. Afterwards, the samples were incubated with primary antibodies for monoclonal mouse IgG2b anti-ChemR23 (1:100, Abcam ab167097), polyclonal rabbit anti-FPR2 (1:50, Abcam ab203129) and monoclonal mouse IgG1 anti-CD45 (1:100, Dako M0701) in antibody diluent (Dako) at 4 °C overnight. After washing in TBS with 0.05% Tween 20, a secondary antibody mix was applied containing 1:400 anti-rabbit Alexa647, 1:500 anti-mouse IgG1 Alexa 555 and anti-mouse IgG2b Alexa 488 in TBS with 10% NGS/1% BSA and incubated for 1 h at room temperature. To reduce autofluorescence, a Vector^®^ TrueVIEW Autofluorescence-Quenching Kit (Vector Laboratories, Maravai LifeScience, Newark, CA, USA) was used for 5 min at room temperature. Counter-staining was performed with 4′,6-Diamidino-2-phenylindol (DAPI, 1:1000) for 15 min at room temperature, and the final slides were covered using Fluoromount™ (Southern Biotech, Birmingham, AL, USA). Multiplex images were taken with a Leica DM6B Thunder microscope (Leica Microsystems, Wetzlar, Germany). The evaluation of the multiplex images was performed with a self-designed macro in ImageJ. The macro assisted in the creation of masks via thresholding, representing positive areas in the four channels DAPI, ChemR23, FPR2, and CD45. Tools such as manual thresholding, denoising and watershed were used to enhance the accuracy of these masks. Once the masks were created, the macro cut and processed them to generate results for single and overlapping areas. The results are related to 100% of the positive staining area. This was achieved by prioritization, which allowed the areas to be cut with a transparent stack of foils. The resulting staining areas and area fractions were used for subsequent calculations of the percentages of ChemR23, FPR2 and CD45 single-, double- and triple-positive stains.

### 2.4. Flow Cytometric Analysis of SPM Receptors and Immune Cell Subsets in Bursa Tissue

A multiplex flow cytometric analysis was performed to gain a deeper insight into the relevant immune cells and non-immune cells in bursa tissue and to understand which cell types express the SPM receptors ChemR23 and FPR2.

For this, bursa tissue was minced and digested for 2 h at 37 °C under constant movement using 0.3% collagenase type CLS II (Biomol) in phosphate-buffered saline (PBS) with Ca^2+^/Mg^2+^. Subsequently, cells were centrifuged, washed and suspended in bursa cryo-medium: DMEM low glucose, 1% Penicillin/Streptomycin (P/S), 20% fetal calf serum (FCS) Superior and 10% dimethylsulfoxide (DMSO) (all Sigma-Aldrich, St. Louis, MO, USA) and stored at −170 °C in liquid nitrogen. For the flow cytometric analysis, cryo-preserved bursa cell isolates were thawed, washed with nuclease medium (47.5% DMEM low glucose, 47.5% RPMI, 5% FCS, 0.02% Universal Nuclease (25 kU, Thermo Scientific, Darmstadt, Germany)) to counteract cell death and stained first with a Zombie UV™ Fixable Viability Kit (BioLegend, San Diego, CA, USA). For Fc receptor blocking, cells were washed with FACS buffer (PBS with 2% FCS, 0.1% sodium azide) and incubated with 5 µL Human TruStain FcX™ (BioLegend) in FACS buffer for 10 min at 4 °C in the dark. Subsequently, cells were incubated with antibodies for the extracellular markers CD3, CD4, CD8, CD31, CD45, CD56, CD80, CD90, CD206, ChemR23 and FPR2 in FACS buffer for 15 min at room temperature in the dark (Table 3). Afterwards, samples were permeabilized for 30 min using the FoxP3 Staining Buffer Set (Miltenyi Biotech, Bergisch Gladbach, Germany) as a precondition for the intracellular staining of Ki-67 and CD68, which lasted 30 min at 4 °C in the dark. Stained cells were measured using a Cytoflex LX System (Beckman Coulter, Brea, CA, USA) and CytExpert software. Fluorescence-minus-one (FMO) controls were used for proper gating. Staining was analyzed using FlowJo v10.9.0. Cells in the bursa were divided into CD45− cells and CD45+ leukocytes. CD45− cells were analyzed for their expression of CD31 (endothelial cells) and CD90 (fibroblasts). Leukocytes were analyzed for their expression of CD68 (macrophages) with characteristic M1 (CD80+) and M2 (CD206+) phenotype markers, CD56 (NK cells) and T cells (CD3+). Furthermore, the SPM receptor expression of ChemR23 and FPR2, as well as proliferation status (Ki-67 expression), was analyzed on these cells. All cell levels are presented as percentages to the respective parent population.

### 2.5. Characterization of Factors Released from Bursa Tissue

To understand the bursa as a reservoir for pro-inflammatory and pro-resolving mediators that might interact with the adjacent rotator cuff tendons, factors released from the bursa samples in tissue culture were characterized.

After bursa harvest, tissue was weighted, placed in a tissue culture dish and incubated with bursa culture medium (DMEM low glucose with 10% FCS Superior and 1% P/S; all Biochrom, Berlin, Germany) at a 1:100 ratio (mg tissue/µL medium) for 3 days at 37 °C. The weight of the bursa samples ranged from 7.2 to 19.7 mg, with a mean of 14.7 mg. PBS was pipetted into the outer ring of the tissue culture dish to avoid liquid evaporation. Thereafter, the tissue culture supernatant was collected, aliquoted and stored at −80 °C until further use. The bursa tissue was subsequently minced and digested with 0.3% collagenase type CLS II solution as described in Section 2.4. The isolated bursa cells were used for mechanical stimulation experiments (see Section 2.7). In the tissue supernatants, the release of the pro-resolving mediators ANXA1, LXA4, RvD1 and RvD2 was investigated using ELISA (ANXA1: Sigma Aldrich RAB1268; LXA4: Cayman Cay590410; RvD1: Cayman Cay500380; RvD2: Cayman Cay501120) according to the manufacturer’s instructions. To investigate the release of further pro- and anti-inflammatory pathway mediators, a Magnetic Luminex^®^ Discovery Assay with 16 analytes was performed (MCP-1, CXCL11, IFN-γ, IL-1β, IL-1ra, IL-6, IL-8, IL-10, IL-13, IL-17/17A, IL-6 Receptor, MMP-1, MMP-2, MMP-3, TIMP-1, TNF-α; R&D Systems/Bio-Techne, Minneapolis, MN, USA; LXSAHM) according to the manufacturer’s manual and measured with a Luminex^®^ 200 analyzer (R&D Systems, Minneapolis, MN, USA). The concentrations of MMP-1 and MMP-3 were out of range and further analyzed using MMP-1 and MMP-3 DuoSet^®^ ELISA kits (MMP-1: DY901B, MMP-3: DY513; both R&D Systems/Bio-Techne).

### 2.6. Assay to Analyze the Effect of Bursa-Released Factors on Tenocytes

In the next step, the effect of bursa-released factors on tenocyte properties, such as cell proliferation, cell migration, gene expression and collagen type I secretion, were investigated. With this, we aimed to gain knowledge on the possible effect of soluble factors released from the native bursa that may regulate tendon healing processes.

For stimulation, two groups of supernatants were chosen, one with high concentrations of pro-resolving mediators and one with low concentrations of pro-resolving mediators, as determined in the characterization of released factors from bursa tissue (Section 2.5). ANXA1 and RvD1 showed the strongest regulation with different severities of rotator cuff disease and were selected for further investigation. A pool of isolated tenocytes (isolation protocol performed as reported previously [35]) from 5 patients with SSP tears was used at passage 2. Tenocytes were seeded with 1.7 × 10^5^ cells/mL into 2-well ibidi^®^ silicone inserts (Ibidi, Gräfelfing, Germany) for a migration analysis, with a growth area of 0.22 mm^2^ and a volume of 70 µL per well, and cultured until confluence. To investigate the possible pro-resolving effect of bursa-released factors, tenocytes were pre-stimulated with 3 ng/mL IL-1β (Peprotech, London, UK; 200-01B) to induce a pro-inflammatory response. After 24 h, IL-1β supernatants and ibidi^®^ inserts were removed, and tenocytes were incubated with the bursa-released supernatants of the respective groups in tenocyte culture medium (DMEM/Hams F-12; 10% FCS, 1% P/S) at a 1:1 ratio. As controls, tenocytes in culture medium alone with/without IL-1β pre-stimulation were used. All stimulations and controls were performed in duplicate. Cell migration was documented after 24 h and 48 h relative to the 0 h time point. After 48 h, the supernatant was removed, aliquoted and stored at −20 °C until further use. Subsequently, a PrestoBlue™ Assay (Invitrogen, Carlsbad, CA, USA) was performed to investigate cell viability according to the manufacturer’s instructions. Afterwards, RNA was isolated from the tenocytes as a pool of the duplicates using a NucleoSpin^®^ RNA Kit, and the gene expression determined using real-time PCR as described in Section 2.2. Collagen type I secretion was analyzed with Pro-Col1a1 DuoSet^®^ ELISA (Bio-Techne DY6220).

### 2.7. Analysis of the Role of Loading on Pro-Resolving and Inflammatory Pathways in Bursa Cells

As the subacromial bursa is a highly mechanically loaded tissue, different strain magnitudes may have a different effect on pro-inflammatory and pro-resolving processes in bursa tissue and, thus, may lead to a different regulation of tendon disease progression or healing.

Bursa-derived cells were isolated from the bursa tissue of the severe disease group from the release experiments 3 days after tissue culture. To investigate the impact of mechanical stimulation, these bursa-derived cells were subjected to physiological (2%) and pathological (8%) uniaxial cyclic loading in collagen-I-coated flexible silicon dishes (Elastosil^®^ M 4641, ratio components A and B 10:1, Wacker, Munich, Germany) for 4 h/day on three consecutive days as described previously [13]. The used stimulation device was adapted from a device developed by Neidlinger Wilke et al. [36]. With the loading device, it is possible to stretch six silicon dishes in parallel with a surface area of 2 × 3 cm. Strain magnitudes between 1 and 8% stretch and strain frequencies of 0.5–2 Hz could be adjusted using a motor (ASB42C048060-ENM, Nanotec Electronic, Feldenkirchen, Germany) and an engine control (CANopen C5-E-1-09, Nanotec Electronic) unit. The strain levels in the bursa in vivo are unknown and adjusted according to knowledge of the tendon, with a strain of up to 5% representing physiological conditions, whereas an 8% strain or higher leads to tissue rupture [37]. Compared to a previous study [13], a stiffer and more tear-resistant silicon was used that resulted in slightly lower strain values, as measured in a speckle analysis. As a positive control, bursa-derived cells were stimulated with 100 ng/mL lipopolysaccharide (LPS; Sigma Aldrich) for the same time period. LPS, as a pathogen-associated molecular pattern (PAMP), was used as strong stimuli to verify that bursa-derived cells are immuno-responsive and that SPM receptor expression can be altered in this cell type. Mechanical loading and LPS stimulation were performed in triplicate, and 1 h after stimulation, one dish was used for RNA isolation and two dishes for analyzing marker expression via flow cytometry. Preliminary experiments defined 1 h after stimulation as optimal time point for RNA isolation. Using flow cytometry, the expression levels of SPM receptors (FPR2, ChemR23), fibroblast markers (CD90, CD105), adhesion molecules (CD54, CD106), human leucocyte antigens (HLA-DR, HLA-ABC) and proliferation marker Ki-67 were analyzed as described in Section 2.4. and according to Table 3. Adhesion molecules and HLAs are important markers for immune recognition and antigen presentation, and they were analyzed to prove the immune-responsiveness of the bursa-derived cells. Furthermore, the gene expression levels of SPM signaling genes (ANXA1, FPR2, ChemR23, GPR18), Col1A1, matrix-degrading enzymes and inhibitors (MMP-1, -2, TIMP-1) and pro-inflammatory cytokines (IL-6, IL-1β) were analyzed using real-time PCR relative to the reference gene HPRT. The protein secretion of ANXA1 and RvD1 in cell culture supernatants was investigated using ELISA (see Section 2.5).

### 2.8. Statistics

A statistical analysis was performed using GraphPad Prism version 7.0.0. The Kruskal–Wallis Test with Dunn’s Multiple Comparison was used to investigate significant differences between the three disease groups and the different cell populations in the bursa tissue (flow cytometry), as well as the different mechanical loading groups. For the stimulation of tenocytes with the bursa supernatants, two sets of statistical comparisons were performed: all groups were compared to either the untreated control or the IL-1β-pre-stimulated group. The sample size varied depending on the availability of bursa samples, and it is indicated in the respective figure legends. The significance was set at *p* ≤ 0.05, and one to three stars indicate the level of significance ranging from *p* ≤ 0.05 to *p* ≤ 0.0001. Additionally, a *p*-value of *p* < 0.1 was used to indicate trends. For qRT-PCR data and protein release, a cluster analysis for multivariate data was performed using the free web tool ClustVis (https://biit.cs.ut.ee/clustvis/ (accessed on 24 August 2023)). The individual values were collapsed as median of each group. Rows (genes or proteins) were centered, and unit variance scaling was applied to rows. Rows were clustered using maximum distance and average linkage.

## 3. Results

### 3.1. Identification of Pro-Resolving and Inflammatory Mediators in Bursae from Patients with Rotator Cuff Disease

Regarding the gene expression level, pro- and anti-inflammatory pathway targets were analyzed in bursae from patients with an intact rotator cuff compared to patients with moderate and severe rotator cuff disease. The cluster analysis revealed a more pronounced expression of pro-inflammatory markers of the IFN-γ and NF-κB pathways in the bursae of intact rotator cuffs, whereas the bursae of moderate and severe disease showed a higher expression of anti-inflammatory pathway targets of the SPM, STAT6 and glucocorticoid pathways (Figure 2A). In a group comparison, significantly downregulated gene expression of interferon regulatory factor 1 (IRF1), IL-6, IL-8 and IL-1β was observed for the bursae of severe and/or moderate disease compared to the intact rotator cuff controls. Additionally, the expression of the pro-inflammatory cytokine IL-1β was decreased in the bursae of moderate disease compared to in the severe stage (Figure 2B).

Regarding the protein level, multiplex immunofluorescence staining revealed FPR2- and ChemR23-positive cells in the perivascular, fibrous or fatty tissue of the subacromial bursa. Single- or double-positive FPR2/ChemR23 cells were partially identified as not only CD45+ leukocytes but also as CD45− cells (Figure 3A). A quantitative analysis using a self-designed image analysis tool in Image J showed no significant differences regarding the quantification of the immunofluorescence staining between the three disease groups. In general, the severe disease group showed a slightly higher CD45 signal, whereas the bursae of moderate stages showed higher total ChemR23 and FPR2 values (Figure 3B). Regarding the distribution of immunofluorescence staining, the majority of the tissue area was single-positive for ChemR23, FPR2 or CD45. The intact controls showed the highest values for double- or triple-positive staining (Figure 3C).

Flow cytometry allowed for a more detailed analysis of the distribution of the different cell types in the bursae and their expression of the SPM receptors ChemR23 and FPR2. Within the immune cell fraction (CD45+), the bursae contained the highest amounts of macrophages (CD68+), followed by T cells (CD3+) and a few NK cells (CD56+) (Figure 4A). Within the non-immune cells (CD45−), fibroblasts (CD90+) represented the dominant cell population (Figure 4B). None of the cell types differed significantly between the disease groups (Figure 4C–F). ChemR23 was expressed mainly on bursa fibroblasts and to a lower extent on endothelial cells (CD31+) and macrophages, whereas a negligible level of T cells expressed ChemR23 (Figure 4G). FPR2 was expressed by a small portion of endothelial cells, followed by an even lower level of macrophages. T cells and fibroblasts seemed to not express FPR2 (Figure 4H). The amount of cells that were double-positive for ChemR23 and FPR2 was low in all investigated cell populations, and their levels did not differ between the cell types (Figure 4I). Regarding the disease groups, the levels of ChemR23+ and ChemR23/FPR2 double-positive macrophages were found to be lower in the bursae of patients with severe disease than in the intact controls or moderate disease, respectively (Figure 4J–K). CD80+ macrophages, which indicate the M1 macrophage phenotype, decreased with severe disease compared to the intact control (Figure 4L), whereas CD206+ M2 macrophages did not show any differences between the disease groups.

### 3.2. Release of SPMs and Inflammatory Mediators from Bursa Tissue

To determine whether SPMs and inflammatory mediators are released in different amounts according to the underlying rotator cuff disease, bursa tissue was incubated for 3 days in culture medium. Afterwards, culture supernatants were analyzed using ELISA and a Luminex^®^ multiplex assay for the protein levels of pro- and anti-inflammatory pathway mediators and SPMs. The cluster analysis revealed an increase in the SPMs LXA4 and RvD2 in severe disease, whereas pro-inflammatory factors (particularly IL-13, IFN-γ, TNF-α, IL-1β, IL-17 and IL-6), matrix remodeling enzymes and inhibitors (MMP-2, MMP-3 and TIMP-1) were enhanced in the bursae of moderate diseases (Figure 5A, and Appendix A). The differences did not reach significance in group comparisons. The bursae of moderate rotator cuff disease showed a trend for an increased ANXA1 release compared to the bursae of the intact rotator cuffs (*p* = 0.099). The bursae of severe diseases similarly released higher amounts of ANXA1. RvD1 release was not detectable in most bursae of the intact rotator cuffs and increased with severe disease, without reaching a significant difference (*p* = 0.055). The release of LXA4 and RvD2 was similarly higher in severe disease than in the intact controls (Figure 5B). All inflammatory markers highly positively correlated with each other (R^2^ > 0.65) or to a lower degree with ANXA1 and LXA4 (R^2^ > 0.32), but no correlations were found between these factors and the resolvins RvD1 and RvD2.

### 3.3. Influence of Bursa-Released Factors on Tenocytes

To investigate the pro-resolving potential of the bursa-released factors, inflamed tenocytes were stimulated with the bursa supernatants of patients with severe rotator cuff disease. Therefore, the bursa supernatants were divided into low or high ANXA1 or low or high RvD1/2 groups according to the previous protein quantification and applied to IL-1β-pre-stimulated tenocytes (Figure 6A). ANX and RvD were chosen as the proteins of interest, as for these two pro-resolving mediators, the strongest regulations between the disease groups were observed, and, thus, they might have a high clinical relevance. In addition to the expected significant difference in the ANXA1 concentration, the high and low ANXA1 supernatants also differed regarding almost all investigated pro- and anti-inflammatory factors, MMPs and TIMPs but not regarding the SPMs LXA4, RvD1 and RvD2. In contrast, the low versus high RvD supernatants exclusively differed in the protein concentrations of RvD1 and RvD2 (Figure 6B). The addition of the bursa supernatants to IL-1β-challenged tenocytes increased cell viability, particularly in both RvD groups (Figure 6C). IL-1β stimulation reduced the migratory potential of the tenocytes, and the bursa supernatants were able to partially reverse the effect (Figure 6D). Col I secretion and Col1A1 expression were decreased in all IL-1β-stimulated tenocytes, and even more pronounced reductions occurred with the addition of the bursa supernatants (Figure 6E,F). The expression of Col3A1 was upregulated in the tenocytes stimulated with IL-1β, and the effect was reversed by the bursa supernatants (Figure 6G). The strong decrease in Col1A1 expression and increase in Col3A1 expression resulted in a lower Col1A1/Col3A1 ratio for all IL-1β-challenged tenocytes than for the unstimulated controls and a further slight decrease with the addition of the bursa supernatants (Figure 6H). ANXA1 and ChemR23 expression was downregulated in tenocytes via IL-1β stimulation and even further reduced by the addition of the bursa supernatants (Figure 6I,J). The expression of the SPM receptor GPR18 was increased with IL-1β stimulation compared to the untreated control, whereas particularly high ANX and RvD supernatants were able to reverse the effect (Figure 6K). IL-1β stimulation induced a pro-inflammatory response in tenocytes via the upregulation of IL-6 and IL-1β expression with slightly, but non-significantly, increased effects with the addition of the bursa supernatants (Figure 6L,M).

### 3.4. Role of Mechanical Loading on SPM Signaling Mediators in Bursa Cells

To understand the regulatory mechanisms of SPM signaling in bursa tissue, the cells from patients with severe rotator cuff disease (n = 6) were subjected to physiological (2%) or pathological (8%) straining under uniaxial cyclic loading for 4 h per day on three consecutive days. Bursa cells are immuno-responsive and reacted on inflammatory stimuli of 100 ng/mL LPS, which served as a positive control, with the upregulation of the SPM receptor FPR2, proliferation marker Ki-67 and adhesion markers CD54 and CD106, as well as the human leucocyte antigens HLA-DR and HLA-ABC. However, neither physiological (2%) nor pathological mechanical straining (8%) alone resulted in the significant regulation of surface markers (Figure 7A and Appendix A). Pathological, but not physiological, mechanical straining increased the gene expression of the SPM receptor ChemR23 and MMP-1 compared to the unstimulated controls. Col1A1 gene expression was slightly decreased (*p* = 0.054), and MMP-2 and TIMP-1 expression increased (*p* = 0.083 and *p* = 0.054, respectively) (Figure 7B). Furthermore, pathological straining increased the secretion of the pro-resolving mediator ANXA1 from bursa cells to the culture medium without reaching a significant difference (*p* = 0.054), whereas RvD1 secretion was not affected (Figure 7C).

## 4. Discussion

The overall aim of this study was to clarify the role of the subacromial bursa in different severities of rotator cuff disease, ranging from an intact rotator cuff to moderate and severe rotator cuff disease. We especially strived to understand the impact of the bursa on the resolution of inflammation in tendon diseases, with a special focus on SPMs. So far, SPMs have not been studied in bursa tissue. Furthermore, the literature on inflammatory cell subsets and connected pro- and anti-inflammatory pathway components in the bursa are limited, and the findings are inconsistent. According to the close localization of tendons and bursae, we hypothesized that the bursa affects inflammation and might release SPMs to counteract tendinopathic conditions in the tendon. In our approach, we (1) identified SPM signaling mediators, immune cells and inflammatory pathway components in bursa tissues; (2) investigated the pro-resolving effect of bursa-released factors on tenocytes; and (3) tackled the role of loading on SPM signaling mediators in bursa cells to understand the regulatory mechanism.

### 4.1. Identification of Pro-Resolving and Inflammatory Mediators in Bursa Tissues

Firstly, we characterized the pro-resolving pathway mediators and the balance between pro-and anti-inflammatory factors in the bursa tissue of patients with different stages of rotator cuff disease, and we showed, for the first time, the presence and possible regulatory potential of SPM signaling mediators in bursa tissue. The bursae from patients with moderate rotator cuff disease had the highest gene expression of factors associated with SPM signaling, such as GPR18 and ANXA1, while pro-inflammatory factors IRF1, IL-6, IL-8 and IL-1β were reduced (Figure 2A,B). Interestingly, in the bursae from patients with severe rotator cuff disease, pro-resolving factors, such as FPR2 and ChemR23, as well as anti-inflammatory mediators, such as CD206, IL-10 and CD1D, prevailed, and pro-inflammatory IRF1 was downregulated compared to the intact group (Figure 2A,B). This partially contradicts the findings of others, showing a rather increased expression of pro-inflammatory factors like SDF1, IL-1β, TNF-α or IL-6 in the bursae of patients with a rotator cuff tear or frozen shoulders [16,17,38]. In addition to the reduced expression of pro-inflammatory factors, the frequencies of pro-inflammatory M1 macrophages was reduced in the bursae of severe disease rotator cuffs (Figure 4L). SPMs act by inhibiting the migration of M1 macrophages into tissues and preventing the secretion of pro-inflammatory cytokines by immune cells [22,23]. Therefore, the reduced pro-inflammatory phenotype in the rotator cuff pathology groups might result from the increased release of pro-resolving mediators that occurred in the bursae of patients with rotator cuff disease (Figure 5B), indicating a counter-regulation of pro-inflammatory processes. When comparing moderate and severe rotator cuff disease, the expected increase in the expression of pro-inflammatory factors such as IL-6 and IL-1β became evident (Figure 2B). These mediators drive, amongst others, tendinopathy [39,40], and once released from the bursa tissue, they can contribute to the chronicity and severity of adjacent rotator cuff tears. However, it is still unclear whether the bursa is the driver of pathological processes in rotator cuff tendinopathy or whether it only reacts to pre-existing pathological changes in the tendons.

The analysis of the SPM receptors FPR2 and ChemR23 using flow cytometry revealed decreased frequencies of ChemR23+ and ChemR23+FPR2+ macrophages in the bursae from patients with severe rotator cuff disease compared to the intact controls or moderate disease. This supports comparable findings regarding the rotator cuff tendons [26], underlining possible linked mechanisms of the two adjacent tissues. Macrophages comprise the dominant immune cell population in the bursa, as reported previously, primarily existing in pathological conditions, but also in healthy conditions [15]. Furthermore, as macrophages and soluble factors are able to modulate inflammatory processes in tenocytes [41], they might be key regulators in the tenocyte immune response. The reduced levels of pro-resolving ChemR23+ and ChemR23+FPR2+ macrophages may indicate a reduced potential to overcome pro-inflammatory processes in the bursa tissue, resulting in worsening symptoms or a lack of contribution to restoring the tissue integrity of the adjacent tendons in patients with severe rotator cuff disease. With further speculation on the interplay between bursae and tendons, the enhanced release of SPMs (Figure 5), with simultaneously unchanged or even reduced SPM receptor expression (Figure 3 and Figure 4) in the bursae itself, might indicate the possible role of bursa tissue in regulating the inflammatory processes of adjacent rotator cuff tendons. Particularly early-stage tendinopathies showed increased amounts of SPM receptors in the tendon [26], which could enable an increased response of the tendon to SPMs released by the adjacent bursa. We showed that SPMs are actors in the bursa and could potentially contribute to tendon regeneration and the restoration of tendon integrity. However, the current findings imply that the endogenous levels of SPMs may be too low to counteract inflammation or boost the regenerative cascade on the tendon rupture side. Therefore, the application of SPMs to the bursa, which could serve as a release reservoir, could have promising potential as a targeted therapeutic intervention to improve tendon healing.

### 4.2. Effect of Bursa-Released Factors on IL-1β-Challenged Tenocytes

With the quantification of further mediators released from the bursa tissue, it became clear that a large variety of soluble pro- and anti-inflammatory mediators could potentially be involved in the biological processes of the bursa itself or adjacent tendon tissue. With the large variety of factors in mind, we aimed to examine the effect of bursa-released factors on the modulation of the response of tenocytes to inflammatory stimuli in the next step of the study. Tenocytes from severe rotator cuff disease were used. These tenocytes did not show a primed pro-inflammatory stage (low IL-6 and IL-1β expression) and were additionally challenged with IL-1β in vitro, which is known as a pro-inflammatory cytokine that is linked to pathological conditions in rotator cuff tendons [39,40,42,43]. IL-1β stimulation led to the expected increase in the pro-inflammatory response of the tenocytes, with the upregulation of IL-6 and IL-1β expression and the downregulation of ANXA1 and ChemR23 expression, as well as a decreased migratory potential and ECM synthesis (Figure 6C–M). This supports previous findings showing that IL-1β modulated the cytokine release of rotator cuff tenocytes or reduced collagen expression [41,44]. The addition of bursa supernatants to IL-1β-pre-treated tenocytes resulted in improved cell viability and migration potential (Figure 6C,D). Despite the well-known function of SPMs in inhibiting the growth and migration of inflammatory cells, it could also be shown that SPMs, such as lipoxins and resolvins, induce the growth of tissue-resident cells like epithelial cells or stem cells [45,46]. On the other hand, ANXA1 is primarily involved in the inhibition of cell proliferation [47]. As it is known that bursae contain high amounts of growth factors, which were not investigated in the present study, the increase in cell proliferation and migratory potential is most likely due to growth factors. As a further effect, bursa supernatants reversed the increased Col3A1 and GPR18 expression mediated by IL-1β treatment to the level of the untreated control cells (Figure 6G,K). In contrast, Col1A1 expression was decreased by IL-1β treatment, and a slightly stronger effect was observed with the addition of bursa supernatants, resulting in a Col1A1/Col3A1 ratio similar to that of IL-1β treatment alone (Figure 6F,H). Collagen type 3 is linked to a lower biomechanical competence of tendons [48]. The decrease in Col3A1 expression by the addition of the bursa supernatants to IL-1β-challenged bursa cells could account for a positive effect on the tendon integrity in the healing process in vivo. Taken together, we conclude that the bursae of diseased patients release a composition of soluble factors able to alter tendon healing processes. Similarly, other studies indicated that the bursa could play a supportive role in tendon healing. The fibrovascular network and growth factor content, the large stem cell potential and biomechanical properties of the bursa are supposed to promote tendon healing [4,10,14,49]. Additionally, it was shown that activated bursa tissue, activated by either tenotomy in vivo or IL-1β stimulation in vitro, is able to direct the healing response in rotator cuff disease via immunomodulation [50]. With this study, we add that soluble factors released from bursa tissue can positively regulate tenocyte properties and might play an important role in the bursa–tendon interplay in vivo.

In the broad range of soluble proteins released from the bursa tissue, we were not able to determine which ones had the highest impact on tenocytes. However, we chose to further investigate the effect of ANXA1 and RvD1 on tenocytes because these two pro-resolving factors showed the strongest regulation between the bursae of intact rotator cuff and moderate and severe disease groups. This might indicate a strong clinical relevance of these pro-resolving mediators in the bursa–tendon interplay. In comparison, stimulation with high versus low ANXA1 and RvD bursa supernatants resulted in no distinctly different responses of the tenocytes. However, slightly stronger differences emerged between the high ANX and low ANX groups, with the high ANX bursa supernatants affecting tenocytes to a larger extent (Figure 6). This could be explained not only by the higher ANXA1 concentration but also by the other pro- and anti-inflammatory factors and MMPs/TIMPs that were increased in these bursa supernatants. In contrast, high versus low RvD bursa supernatants only differed in RvD1 and RvD2 concentrations, whereas the other quantified factors showed similar concentrations (Figure 6B). MMPs and TIMPs were investigated as factors that are not only important for tissue remodeling but that also play an immunomodulatory role and are involved in the disease progression of rotator cuff pathology [40,41]. MMPs and TIMPs were released in high amounts from the bursa tissue of all rotator cuff disease stages and could have had a strong impact on the cellular processes of the tenocytes. Regarding the pro-resolving effect of SPMs on tenocytes, other studies showed that LXB4, LXA4, RvE1 and Maresin 1 (MaR1) prevented the pro-inflammatory response to the IL-1β stimulation of tendon stromal cells from the Achilles tendon or rotator cuff [28,51]. The different study designs with IL-1β stimulation after the SPM incubation point to a different mechanism of the protection of tendon cells from pro-inflammatory stimuli rather than the resolution of an existing pro-inflammatory state. Pre-stimulation with IL-1β led to a strong pro-inflammatory response in tenocytes, which the pro-resolving mediators in the bursa supernatants could not counteract. Additionally, the high levels of pro-inflammatory factors in the bursa supernatants might be more potent to influence tenocyte properties. This could also explain the slightly increased expression of IL-6 and IL-1β in the IL-1β-challenged tenocytes that were stimulated with bursa supernatants (Figure 6L,M). In the case of a strong persistent inflammatory condition, higher initial concentrations of pro-resolving mediators are required to counteract inflammation. Regarding the therapeutic effect of SPMs released from the bursa in the resolution of tendon inflammation, further studies are needed to fully explore the potential benefit.

### 4.3. Effect of Loading on Pro-Resolving Processes in Bursa-Derived Cells

Especially in persisting inflammatory settings, mechanical loading may further promote inflammatory processes in tendons, which was, amongst others, verified by the fact that the use of anti-inflammatory drugs reduces the response in loaded tendons in vivo [52]. In vitro studies revealed that loading or overloading leads to an increased expression of inflammatory markers, to a delayed switch from M1 to M2 macrophages and to reduced levels of regulatory T cells [53,54]. The subacromial bursa experiences high mechanical loads during each movement of the shoulder, and we previously showed that bursa cells are mechano-responsive and react differently to physiological and pathological loading conditions in vitro [13]. Therefore, in the last part of the study, we aimed to investigate the impact of different mechanical loading conditions on bursa cells concerning the regulation of mediators related to the resolution of inflammation. The overall results provide evidence that mechanical loading partially regulates pro-resolving processes in bursa cells. In particular, the pathological loading conditions of 8% strain resulted in an increased expression of the SPM receptor ChemR23 and the slightly upregulated secretion of ANXA1 (Figure 7B,C). Moreover, pathological loading resulted in an altered gene expression of ECM remodeling factors (MMPs) (Figure 7B). The impact of loading was further investigated for the surface marker expression of SPM receptors, as well as fibroblast markers, adhesion molecules and HLAs on bursa cells, using flow cytometry. Mechanical loading alone under the tested conditions did not influence the surface marker expression of isolated bursa cells (Figure 7A and Appendix A), which might indicate that higher strains or a longer loading duration would be necessary. A period of stress of 4 h per day for three consecutive days represents only a brief moment of everyday stress in vivo, and cells isolated from such a heavily stressed tissue may require a stronger trigger. Furthermore, loading alone might not be potent enough to induce a clear pro-resolving response or immune response, but synergistic processes of inflammatory and mechanical triggers are required. FPR2 was not expressed on isolated and cultured bursa cells, whereas ChemR23 was found to be expressed on all bursa cells regardless of being unstimulated or mechanically loaded (Figure 7A), which supports the present findings of the flow cytometric analysis of bursa tissue (Figure 4G,H). Triggering an inflammatory response as a positive control using LPS resulted in the upregulation of FPR2 expression on bursa cells and a slightly decreased ChemR23 expression (Figure 7A and Appendix A). Similarly, ChemR23 gene expression was reduced in tenocytes stimulated with IL-1β (Figure 6J), which indicates comparable pathways in these two cell types. The increase in ChemR23 gene expression but the unchanged ChemR23 surface marker expression under pathological loading might be explained by time-related constraints that possibly resulted in distinct regulations on the surface marker level at later time points after stimulation that were not investigated in the present approach. In summary, SPM signaling mediators in bursa cells are partially altered under pathological loading conditions, suggesting that bursa cells attempt to counter-regulate the inflammatory mechanisms induced by pathological overloading. In the bursa–tendon interplay, mechanical forces play an important role, and the current results indicate that strains that would result in tendon injury in vivo (8% stretch) [37] partially trigger the upregulation of pro-resolving processes in the bursa, which could benefit the adjacent tendon. However, besides the mechanical trigger, the inflammatory and extracellular environment also influences the bursa–tendon interplay in vivo, which was not taken into account in the present approach. Future studies must shed a more detailed light on these synergies.

### 4.4. Limitations

Regarding limitations, the age and sex of the patients could not be considered as independent confounders due to the limited sample size—although it can be expected that these would partially influence the results in the disease group comparison. As a further limitation, it has to be kept in mind that the bursae of the intact rotator cuff group were derived from patients with shoulder pathology, such as traumatic AC joint luxation or severe shoulder instability. Due to their more acute nature, these pathologies may lead to a greater infiltration of immune cells. Furthermore, in the intact rotator cuff group, samples from patients surgically treated with an open procedure were also included, whereas the rotator cuff tear patients were exclusively treated arthroscopically. The more invasive removal of biopsies could have an effect on the results obtained, which might also partially explain the high pro-inflammatory potential of the intact rotator cuff group. However, no distinct differences were observed when comparing the results of the intact rotator cuff samples from open versus arthroscopic surgeries. In the analysis of the influence of bursa-released factors on tenocytes, the study concentrated on pro-inflammatory, anti-inflammatory and pro-resolving factors. As this is most likely just a small fraction of the released soluble factors, other markers such as growth factors could also have had an effect on the tenocytes. A more detailed investigation is needed to further explore this in the future. Furthermore, it is clear that more tissues are involved in the shoulder joint that might affect tendon tissue integrity. For example, a recent study showed increased synovitis in patients with more severe shoulder diseases [55]. Therefore, the results of the present study provide novel insights into the possible interplay of bursae and tendons, but more investigations are necessary to elucidate the complex interplay of tissues in shoulder pathologies.

## 5. Conclusions

In conclusion, our study provides evidence that pathway mediators regulating the resolution of inflammation in bursa tissue are altered depending on the severity of rotator cuff disease. Furthermore, pro-resolving mediators are released from bursa tissue and may be able to modulate inflammatory processes on the tendon rupture side. Overall, our data support the importance of the bursa in the interplay with adjacent rotator cuff tendons. Therefore, the bursa could be a promising tool to boost healing processes in the diseased tendon, either by its own endogenous healing-promoting properties or by the release of soluble promoting factors to the injured tendon side. Here, SPMs could represent promising therapeutics, as they are able to dampen excessive or prolonged pro-inflammatory processes while maintaining important pro-resolving mediators. However, further knowledge is needed to gain a full mechanistic understanding of the bursa–tendon interplay with regard to the interrelationship between inflammation, resolution and mechanical forces.

## Figures and Tables

**Figure 1 cells-13-00017-f001:**
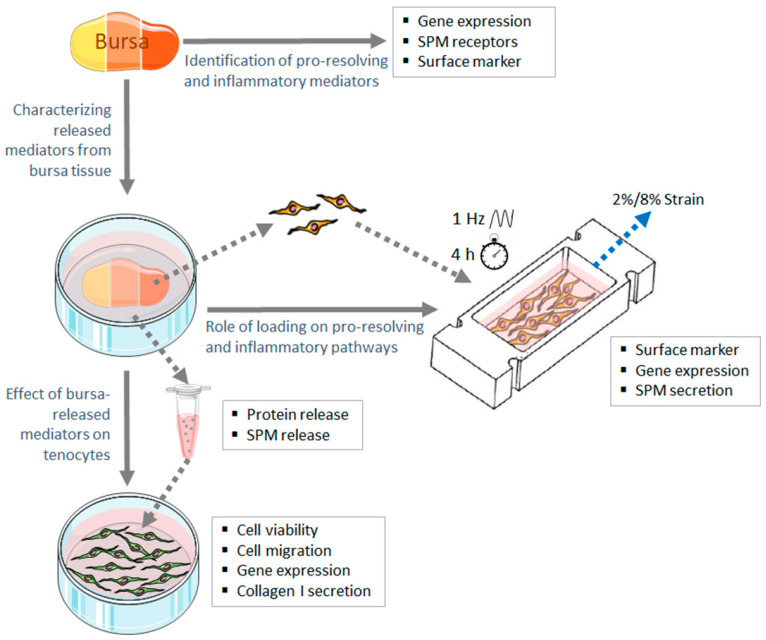
Study design to investigate the role of the subacromial bursa in pro-resolving processes at the human rotator cuff. The yellow-to-red bursa tissue indicates the different disease stages of the rotator cuff. The image was created with graphics from Servier Medical Arts. SPM: specialized pro-resolving mediators.

**Figure 2 cells-13-00017-f002:**
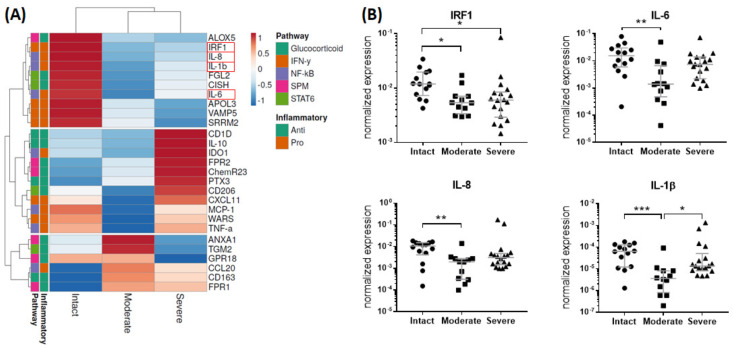
Gene expression of pro- and anti-inflammatory pathway targets in bursae of intact rotator cuff (n = 14), moderate disease (n = 12) or severe disease (n = 18) measured using real-time PCR. (**A**) Heatmap derived from normalized gene expression values by collapsing the individual values as median of each group. Rows (genes) were centered, and unit variance scaling was applied to rows. Rows were clustered using maximum distance and average linkage. (**B**) Scatter plot of selected genes (IRF-1, IL6, IL8, IL1β) with significant differences between groups (highlighted with a red box in (**A**)). Gene expression of target genes was normalized to the three reference genes Cyclophilin A (Ppia), Hypoxanthin-Phosphoribosyl-Transferase (HPRT) and 18s and calculated using an efficiency corrected equation. Dunn’s Multiple Comparison test was conducted, with * *p* < 0.05, ** *p* < 0.01 and *** *p* < 0.001 indicating significant differences. ALOX5: Arachidonat-5-Lipoxygenase; IRF1: interferon regulatory factor 1; IL-8: Interleukin 8; IL-1β: Interleukin 1β; FGL2: fibrinogen-like protein 2; CISH: cytokine inducible SH2-containing protein; IL-6: Interleukin 6; APOL3: Apolipoprotein L 3; VAMP5: vesicle-associated membrane protein 5; SRRM2: serine/arginine repetitive matrix 2; CD1D: Cluster of Differentiation 1D; IL-10: Interleukin 10; IDO1: indoleamine 2,3-dioxygenase 1; FPR2: formyl peptide receptor 2; ChemR23: Chemerin Receptor 23; PTX3: Pentraxin 3; CD206: Cluster of Differentiation 206; CXCL11: C-X-C motif chemokine ligand 11; MCP-1: monocyte chemoattractant protein 1; WARS: tryptophanyl-tRNA synthetase; TNF-α: tumor necrosis factor α; ANXA1: Annexin A1; TGM2: Transglutaminase 2; GPR18: G-protein-coupled receptor 18; CCL20: C-C motif chemokine ligand 20; CD163: Cluster of Differentiation 163; FPR1: formyl peptide receptor 2.

**Figure 3 cells-13-00017-f003:**
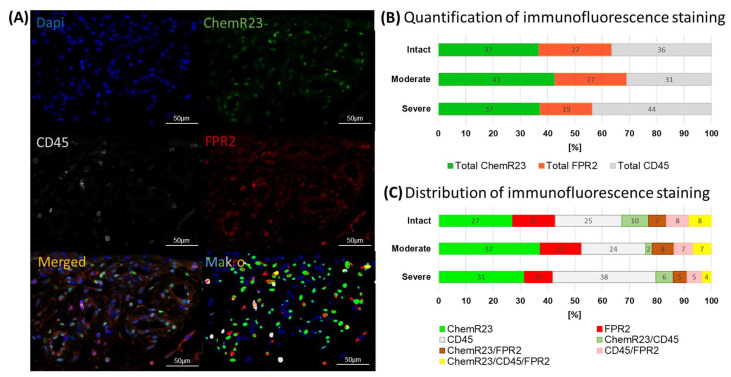
Analysis of SPM receptors ChemR23 and FPR2 in histological sections: (**A**) Exemplary immunofluorescence image of a bursa sample from the severe disease group showing single staining channels for Dapi, ChemR23, FPR2 and CD45, as well as channel overlap (merged), and the respective result after image analysis (Makro). Scale bars represent 50 µm. (**B**) Quantification of immunofluorescence staining of total ChemR23, FPR2 and CD45 (single- + double- + triple-positive staining area) in bursae of intact controls (n = 8), moderate disease (n = 7) and severe disease (n = 8) using a self-designed image analysis tool. (**C**) Distribution of immunofluorescence staining area of single-positive ChemR23, FPR2 and CD45 signal; double-positive ChemR23/FPR2, ChemR23/CD45 and FPR2/CD45 signal; or triple-positive ChemR23/FPR2/CD45 signal. The staining areas are given in % to the total positive staining area.

**Figure 4 cells-13-00017-f004:**
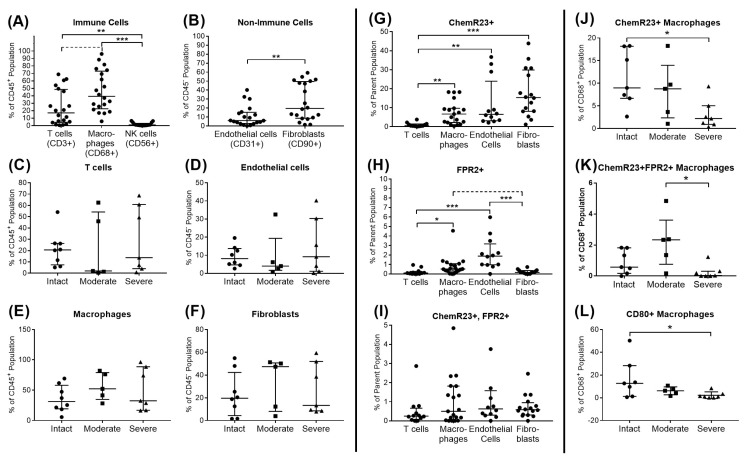
Quantification of cells isolated from bursa tissue using flow cytometric analysis. (**A**) CD45+ immune cells (T cells (CD3+), macrophages (CD68+), NK cells (CD56+)) and (**B**) CD45− non-immune cells (endothelial cells (CD31+), fibroblasts (CD90+)) in all investigated bursae (n = 20) and distribution of (**C**) T cells, (**D**) endothelial cells, (**E**) macrophages and (**F**) fibroblasts between intact (n = 7–8), moderate (n = 5) and severe (n = 6–7) disease given as percentage to CD45+ or CD45− populations. (**G**–**I**) Percentages of (**G**) ChemR23+, (**H**) FPR2+ and (**I**) ChemR23+FPR2+ cells among the main analyzed immune and non-immune cell populations. (**J**–**L**) Distribution of (**J**) ChemR23+ macrophages, (**K**) ChemR23+FPR2+ macrophages and (**L**) CD80+ macrophages between intact, moderate and severe disease. Statistics: Dunn’s Multiple Comparison test and Mann–Whitney U Test given as median with interquartile range. * *p* < 0.05, ** *p* < 0.01, *** *p* < 0.001; a dashed line indicates a trend (*p* < 0.1).

**Figure 5 cells-13-00017-f005:**
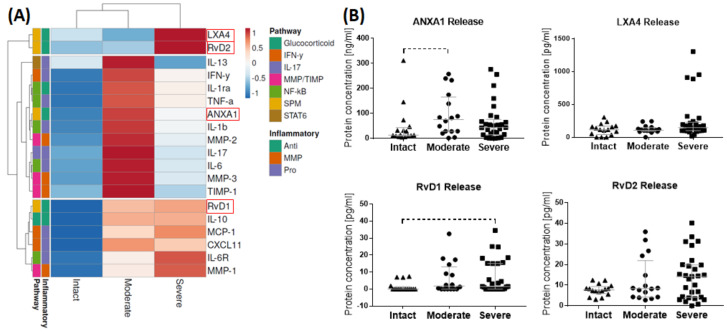
Release of pro-resolving and inflammatory mediators in tissue culture of bursae of patients with intact rotator cuff (n = 16), moderate disease (n = 16) or severe disease (n = 29) measured using ELISA and Luminex multiplex assay. (**A**) Heatmap derived from normalized protein concentrations by collapsing the individual values as median of each group. Rows (proteins) were centered, and unit variance scaling was applied to rows. Rows were clustered using maximum distance and average linkage. (**B**) Scatter plots of release of selected pro-resolving mediators (highlighted with a red box in (**A**)) showing the protein concentrations of ANXA1, LXA4, RvD1 and RvD2 as medians with interquartile ranges. Dunn’s Multiple Comparison test was conducted, with *p* < 0.1 indicating trend for differences with dashed line.

**Figure 6 cells-13-00017-f006:**
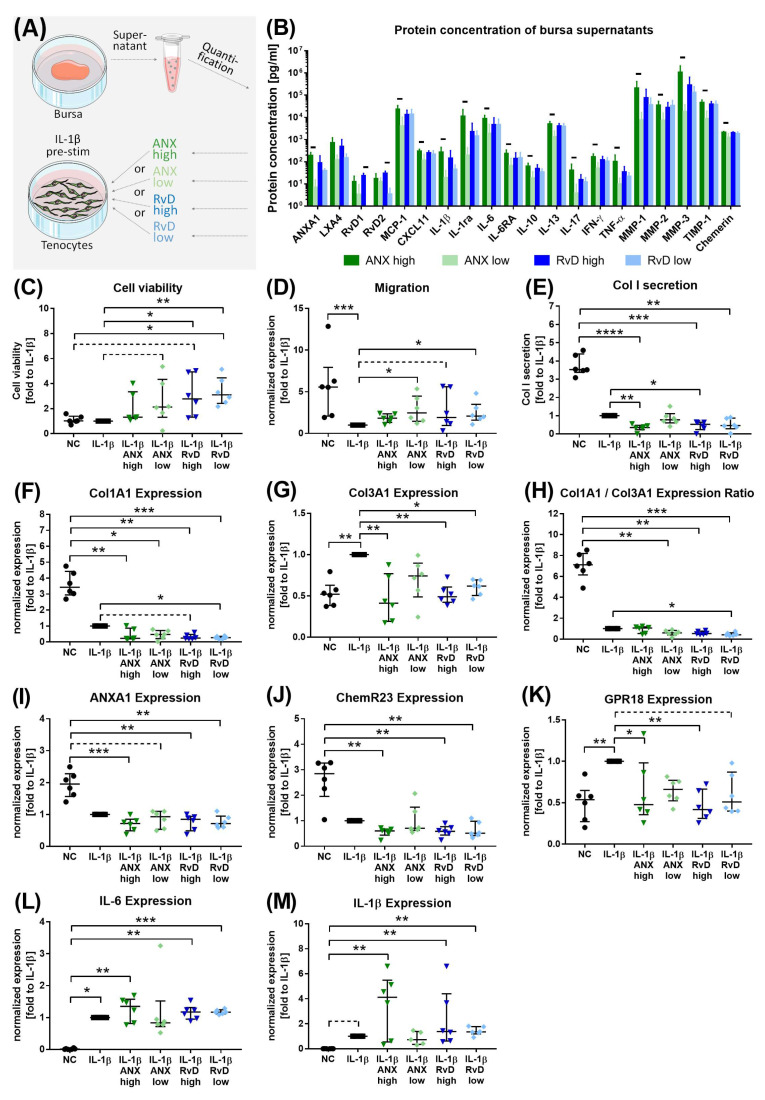
Influence of bursa-released factors on tenocytes: (**A**) Experimental design, (**B**) protein concentrations measured using ELISA or Luminex^®^ Assay in high ANX, low ANX, high RvD and low RvD groups (each n = 6). (**C**–**M**) Results of either unstimulated tenocytes as negative control (NC) or tenocytes pre-stimulated with 3 ng/mL IL-1β 24 h before adding bursa supernatants with high or low ANX or RvD concentrations. Values were normalized to the IL-1β stimulation group (set to 1). (**C**) Cell viability measured using Presto Blue^®^ assay, (**D**) migration analyzed using scratch assay, (**E**) Col I secretion measured using ELISA. (**F**–**M**) Gene expression of target genes (Col1A1, Col3A1, ANXA1, ChemR23, GPR18, IL-6, IL-1β) normalized to the housekeeping gene HPRT, calculated using an efficiency corrected equation and given as fold to the IL-1β stimulation group. Statistics: Dunn’s Multiple Comparison test, * *p* < 0.05, ** *p* < 0.01, *** *p* < 0.001, **** *p* < 0.0001, dashed lines indicate trends (*p* < 0.1).

**Figure 7 cells-13-00017-f007:**
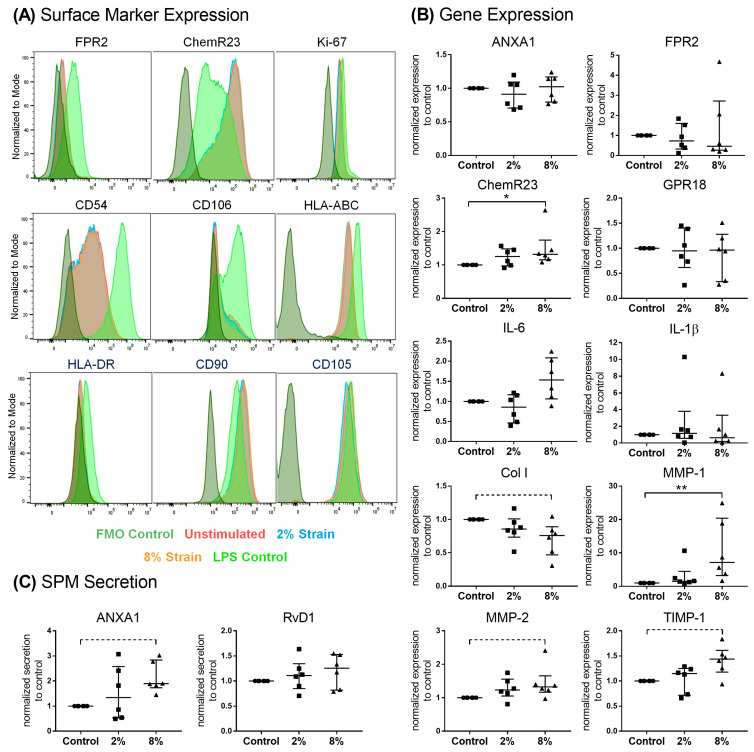
Role of mechanical loading on SPM signaling mediators in bursa cells (n = 6). (**A**) Exemplary results shown as histograms of surface marker expression of unstimulated bursa cells and after mechanical stimulation with 2% and 8% strains. In addition, 100 ng/mL LPS was used as positive control to show the general immune responsiveness of the bursa cells. FMO (fluorescence-minus-one) controls were included for proper gating. (**B**) Gene expression of target genes (ANXA1, FPR2, ChemR23, GPR18, IL-6, IL-1β, Col1A1, MMP-1, MMP-2, TIMP-1) normalized to the reference gene HPRT, calculated using an efficiency corrected equation and given as fold to the unstimulated control. (**C**) Secretion of ANXA1 and RvD1 normalized to unstimulated control (set to 1). Statistics: Dunn’s Multiple Comparison test, * *p* < 0.05, ** *p* < 0.01; a dashed line indicates trends (*p* < 0.1).

**Table 1 cells-13-00017-t001:** Patient demographics.

	Intact Rotator Cuff	Moderate Rotator Cuff Disease	Severe Rotator Cuff Disease
N-number total	28	24	45
Age (Mean ± StD)	32.8 ± 11	53.1 ± 11.8	59.9 ± 8.7
BMI (Mean ± StD)	24.3 ± 3.3	26.6 ± 4.1	27.9 ± 4.9
Female/Male (N)	4/24	6/18	16/29
Disease	Primary shoulder instability: 8 Recurrent shoulder instability: 3 AC joint luxation: 17	Impingement: 2 Partial SSP tear: 22 Intratendinous: 2 Articular side: 7 Bursa side: 13	Full-thickness SSP tear: 45
Classification	-	Snyder: A1: 3, A2: 2, A3: 2 B1: 8, B2: 3, B3: 2	Patte: 1.5 ± 0.7 Bateman: 2.3 ± 0.6

**Table 2 cells-13-00017-t002:** Investigated pro-resolving and pro- and anti-inflammatory markers.

	Abbreviation	Full Name	Function
SPM signaling	ANXA1	Annexin A1 (Target for FPR2)	SPM
	ALOX5	Arachidonat-5-Lipoxygenase	Enzyme for SPM synthesis
	FPR2/ALX	Formyl peptide receptor 2 (Ligands: LXA4, ANXA1, RvD1, RvD2)	Receptors for SPMs
	ChemR23/CMKLR1	Chemerin Receptor 23/Chemokine-like receptor 1 (ligands: RvE1, Chemerin)	
	GPR32	G-protein-coupled receptor 32 (ligand: RvD1)	
	GPR18	G-protein-coupled receptor 18 (ligand: RvD2)	
Pro-/anti-inflammatory anti-inflammatory	CXCL11	C-X-C motif chemokine ligand 11	IFN pathway targets
	IRF1	Interferon regulatory factor 1	
	WARS	Tryptophanyl-tRNA synthetase	
	VAMP5	Vesicle-associated membrane protein 5	
	SRRM2	Serine/arginine repetitive matrix 2	
	APOL3	Apolipoprotein L 3	
	IL-6	Interleukin 6	NF-κB pathway targets
	TNF-α	Tumor necrosis factor α	
	CCL20	C-C motif chemokine ligand 20	
	MCP1	Monocyte chemoattractant protein 1	
	IL-8	Interleukin 8	
	IDO1	Indoleamine 2,3-dioxygenase 1	
	IL-1β	Interleukin 1β	
	TGM2	Transglutaminase 2	STAT-6 pathway targets
	CD206	Mannose receptor	
	CISH	Cytokine inducible SH2-containing protein	
	FGL2	Fibrinogen-like protein 2	
	IL-13	Interleukin 13	
	CD163	CD163 molecule	Glucocorticoid receptor pathway targets
	*IL-10*	Interleukin 10	
	*CD1D*	Cluster of Differentiation 1D	
	*PTX3*	Pentraxin 3, long	

**Table 3 cells-13-00017-t003:** Flow cytometry antibodies.

Target	Fluorochrome	Clone	Manufacturer	Dilution
Live/Dead	Zombi UV	-	BioLegend	1:100
CD3	BV650	OKT3	BioLegend	1:20
CD31	BV605	WM59	BioLegend	1:100
CD45	Pac. Blue	J33	Beckman Coulter	1:40
CD54	PE/Dazzle 594	HA58	BioLegend	1:10
CD56	APC-F700	HCD56	BioLegend	1:20
CD68	PE/CF594	Y1/82A	BD Horizon *	1:50
CD80	BV785	2D10	BioLegend	1:10
CD90	FITC	5E10	BioLegend	1:200
CD105	PE/Cy7	43A3	BioLegend	1:200
CD106	BV421	STA	BioLegend	1:20
CD206	APC-F750	15-2	BioLegend	1:20
ChemR23	APC	15-2	R&D Systems **	1:15
Ki-67	eFluor506	SolA15	eBiosciences ***	1:40
FPR2	PE	304405	R&D Systems	1:15

* BD Horizon, Franklin Lakes, NJ, USA; ** R&D Systems, Minneapolis, MN, USA; *** eBiosciences, San Diego, CA, USA.

## Data Availability

The data that support the findings of this study are available on request from the corresponding author (F.K.).

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
