# Peer review of "Pro-Resolving Mediators in Rotator Cuff Disease: How Is the Bursa Involved?"

_cells, 2023, doi:10.3390/cells13010017_

Round 1

Reviewer 1 Report

Comments and Suggestions for Authors

This study aimed to investigate the production of SPM by bursa tissue of patients with rotator cuff diseases and the production of SPM by bursa-derived cells after mechanical loading. The effects of bursa-derived cells on the resolution of inflammation and functions of IL-1b-treated tenocytes were also investigated. Better understanding of the role of bursa in regulating inflammation in the shoulder would shed light on the treatment of rotator cuff disease. However, the hypothetical roles of bursa in tendon injury and repair and the justification were not clearly presented. This makes it difficult to interpret the results of this study.

My specific comments are listed below for the authors’ information.

Introduction

Line 65-67, clarify if SDF-1 and TGF-b are considered as inflammatory cytokines.

The hypothesis should be presented here. It only appears in the discussion section in this manuscript and was not clear and specific. The rationale and the expected outcome in each experiment were not clear.

Methods

Iine 233-236, the size of the bursa tissue might affect the release of inflammatory or growth factors. How did the authors control the size of the bursa tissue?

The sample sizes for some experiments were missing.

The authors needed to clarify if consent has been obtained from patients before clinical sample collection.

Besides the bursa, did the authors expected any roles of other adjacent tissues in influencing local inflammation and rotator cuff tendon diseases? This should be discussed.

The rationale of using different bursa-derived cells in the mechanical loading experiment and co-cultured experiments should be more clearly justified.

More details about the mechanical loading device and the adaption made should be provided.

Line 289 and line 293, please clarify the loading duration in the experiment.

The statistical analysis section is missing.

In general, this reviewer found that the methodology was not clear and difficult to follow.

Results

3.1, discuss why the expression of pro-inflammatory markers was more pronounced in bursa of intact rotator cuff and the expression of anti-inflammatory markers was higher in bursa of patients with moderate and severe rotator cuff tears.

For the co-culture experiment, it is not clearly why supernatant with high/low ANX and RvD was selected for further analysis. How would other factors in the supernatant influence the outcomes.

Discussion

The results were not well discussed. How is the bursa involved as what the manuscript title suggests?

The weaknesses of the study should be more thoroughly discussed.

Comments on the Quality of English Language

There are some grammatical mistakes. Please check the manuscript.

Reviewer 2 Report

Comments and Suggestions for Authors

The authors explore the influence of the subacromial bursa on neighboring tendon healing in rotator cuff disease, focusing on specialized pro-resolving mediators (SPMs) and inflammatory processes. Investigating SPMs and mechanical loading effects on bursa cells sought to understand their role in modulating inflammation and impacting adjacent rotator cuff tendon recovery, potentially informing targeted therapies for improved tendon healing. This work needs some revisions to be accepted.

1. How did the levels of ChemR23+ and ChemR23/FPR2 double positive macrophages vary among the disease groups, and what implications did these variations suggest in the context of disease severity?

2. What implications do the observed gene expression patterns (upregulation or downregulation of specific markers) have on the understanding of disease progression or potential treatment strategies?

3. How do the heightened levels of IL-6 and IL-1β expression induced by bursa supernatants align with the expected resolution properties of these factors? What implications might this amplified response have on the overall inflammatory resolution process in the tenocytes?

4. Could you elaborate on the implications of the observed discrepancies between the high and low ANXA1 or RvD1/2 groups regarding the expression of various factors, MMPs, TIMPs, and SPMs in the supernatants? What might these differences signify in terms of resolving inflammation or impacting cellular responses?

5. Can you provide insights into the observed increase in cell viability and the restoration of migratory potential upon exposure to bursa supernatants? What mechanisms might be contributing to these effects in the context of the inflammatory challenge posed by IL-1β?

6. What implications can be drawn from the altered Col1A1/Col3A1 ratio in tenocytes stimulated with bursa supernatants? How might this shift in the ratio impact tissue integrity or repair mechanisms in rotator cuff diseases?

7. What were the specific parameters used to subject cells from patients with severe rotator cuff disease to physiological (2%) or pathological (8%) straining? How were these straining levels determined, and why were these percentages chosen?

8. Regarding the findings related to physiological and pathological mechanical straining, what implications can be drawn from the lack of significant regulation of surface markers when applied individually (2% or 8%)? How might these results contribute to understanding the influence of mechanical forces on SPM-related responses?

9. In cases where gene expression changes show a trend toward significance (p=0.054, p=0.083, etc.), what implications do these trends hold for understanding the impact of mechanical straining on SPM signaling and tissue-related factors? How might these trends guide future investigations or hypotheses?

10. Can you explain the rationale behind using 100 ng/mL LPS as a positive control for inflammatory stimuli and how the bursa cells' response to LPS stimulation is relevant to understanding SPM signaling regulation?

11. In the context of the study, why were markers like FPR2, Ki-67, CD54, CD106, HLA-DR, and HLA-ABC chosen for assessment? What significance do the observed upregulations of these markers in response to LPS stimulation hold for understanding the immune responsiveness of bursa cells?

12. How do these findings contribute to the understanding of SPM regulatory mechanisms within bursa tissue under mechanical strain, particularly in the context of rotator cuff disease? What are the potential clinical or therapeutic implications of these observations?

Reviewer 3 Report

Comments and Suggestions for Authors

The manuscript entitled "Pro-resolving mediators in rotator cuff disease: how is the bursa involved?" by Klatte-Schulze et al. is a very well written manuscript. It is easy to read and deals with an extremely controversial topic, as the function of the bursa is still underestimated and its influence on the surrounding tissue is not taken seriously. For this reason, this publication is based on excellent experimental work. The number of patient data included is remarkable. The methods used to answer the research question posed at the beginning are of the highest experimental standard. In summary, it can be said that this is a flawlessly conducted work that stands out not only for its extensive methods, but also for its high donor variability, so that it has managed to get closer to the question of regulating mediators in rotatory cuff diseases. It was possible to show how important the bursa is not only in injuries but also in healing processes. congratulations on this successful work.

Best regards

Round 2

Reviewer 1 Report

Comments and Suggestions for Authors

Most comments are addressed.

Comments on the Quality of English Language

There are still grammatical mistakes in the manuscript.

Reviewer 2 Report

Comments and Suggestions for Authors

Thank you for your responses.